# Radioiodine Exhalation Following Oral I-131 Administration in a Mouse Model

**DOI:** 10.3390/biomedicines13040897

**Published:** 2025-04-08

**Authors:** Klaus Schomäcker, Thomas Fischer, Ferdinand Sudbrock, Daniela Strohe, Sebastian Weber, Beate Zimmermanns, Felix Dietlein, Philipp Krapf, Harald Schicha, Markus Dietlein, Alexander Drzezga

**Affiliations:** 1Department of Nuclear Medicine, Faculty of Medicine and University Hospital Cologne, University of Cologne, Kerpener Str. 62, 50937 Cologne, Germany; thomas.fischer@uk-koeln.de (T.F.); ferdinand.sudbrock@uk-koeln.de (F.S.); beate.zimmermanns@uk-koeln.de (B.Z.); p.krapf@fz-juelich.de (P.K.); harald.schicha@uni-koeln.de (H.S.); markus.dietlein@uk-koeln.de (M.D.);; 2Practice for Gynecology and Prenatal Medicine, Dres. med. Horz-Wilhelm/Strohe, Römerfeld 1, 50129 Bergheim, Germany; danielastrohe@gmx.de; 3Department for Internal Medicine and Intensive Care Medicine, Marien Hospital Düsseldorf GmbH, Rochusstr. 2, 40479 Düsseldorf, Germany; seb.-weber@web.de; 4Computational Health Informatics Program, Boston Children’s Hospital, Harvard Medical School, Boston, MA 02115, USA; felix.dietlein@googlemail.com; 5Forschungszentrum Jülich GmbH, Institute of Neuroscience and Medicine, Nuclear Chemistry (INM-5), Wilhelm-Johnen-Straße, 52428 Jülich, Germany; 6German Center for Neurodegenerative Diseases (DZNE), Bonn-Cologne, Venusberg-Campus 1/99, 53127 Bonn, Germany

**Keywords:** iodine-131, exhalation, radioiodine therapy

## Abstract

**Background**: The exhalation of radioiodine following radioiodine therapy (RIT) presents a challenge in radiation protection, though the mechanisms remain incompletely understood. Previous studies have indicated that radioiodine is predominantly exhaled in an organically bound form in humans. **Methods**: This study investigates the chemical composition and exhaled amounts of radioiodine, as well as the impact of thyroid-targeted pharmacological interventions, using a controlled mouse model. Female Balb/c mice (25 g) were administered oral doses of radioiodine (0.1, 1, 2, 10, and 23 MBq per animal) with and without prior treatment using thyroid-blocking agents (stable iodine, perchlorate) or antithyroid drugs (carbimazole). Exhaled radioiodine was collected in metabolic cages, separating chemical forms (aerosolized iodine, elemental iodine, organically bound iodine), and quantified via scintillation counter. **Results**: The exhaled radioiodine activity was proportional to the administered dose (0.2–0.3%). Thyroid-blocking agents increased exhalation, shifting toward elemental iodine. Antithyroid drugs reduced exhalation but increased aerosol formation, particularly at higher I-131 doses. Organically bound iodine remained the predominant exhaled species in all groups. **Conclusions**: These results highlight the critical role of the thyroid in radioiodine organification. The blockade of thyroid uptake disrupted the formation of organically bound iodine, suggesting that iodine organification requires passage through the thyroid. Additionally, the results support the hypothesis that iodine metabolism outside the thyroid is less efficient, contributing to the formation of organic iodine species. Radical formation is likely a key factor in generating these volatile iodine species, with radiation-induced iodine and methyl radicals playing a role in their formation.

## 1. Introduction

Radioiodine therapy (RIT) is a well-established treatment for various thyroid disorders [1,2,3,4,5,6]. At the core of RIT is the radionuclide [^131^I]I-iodine, a classic theranostic agent that integrates both diagnostic and therapeutic capabilities. Iodine-131 emits therapeutic β-radiation, which selectively ablates thyroid tissue, while simultaneously providing γ-radiation for imaging and treatment monitoring [7,8].

A critical concern following RIT is radiation exposure to healthcare personnel, other patients, and the general public [9,10,11,12]. This exposure arises from two distinct sources: first, gamma radiation emitted directly from the patient’s body [13], leading primarily to external radiation exposure, and second, radioactive iodine exhaled by the patient [14,15,16,17,18], potentially causing incorporation via inhalation by healthcare staff and by-standers.

To evaluate the potential risks associated with airborne radioiodine, our research group previously measured radioactivity levels in ambient air within patient rooms after radioiodine therapy [17] and analyzed exhaled breath samples collected directly from treated patients [18].

Three primary chemical species of exhaled iodine were identified: organically bound iodine, elemental iodine (I_2_), and aerosol-bound iodine. Notably, organically bound iodine was the dominant species shortly after administration, but its proportion decreased over time in favor of elemental and aerosol-bound iodine. Across both studies, approximately 0.01% of the administered radioiodine activity was released into the environment via exhalation.

In ambient air studies, a strong correlation was observed between thyroid iodine uptake and the proportion of exhaled organically bound iodine. Patients with higher thyroidal iodine uptake, such as those with Graves’ disease, exhaled significantly more radioiodine than patients with thyroid carcinoma, whose limited or absent thyroid tissue restricted iodine storage and subsequent exhalation. This correlation supports the hypothesis that iodine organification within the thyroid is a key determinant in the formation of exhalable organic iodine species [17].

However, direct breath measurements from RIT patients revealed a more complex pattern. Although over 94% of exhaled iodine consistently appeared in organically bound form, regardless of the underlying thyroid disease, the correlation between thyroid iodine uptake and the extent of organic iodine exhalation was weaker than expected. Notably, patients with thyroid carcinoma, despite their reduced thyroid function, exhibited longer exhalation half-lives than those with hyperfunctioning thyroids. This unexpected finding suggests that additional factors beyond thyroidal uptake, potentially involving extrathyroidal tissues or systemic metabolic processes, contribute to the formation of exhalable organic iodine species [18].

To resolve these discrepancies and gain deeper insight into the mechanisms underlying iodine organification and exhalation, the present mouse study was designed using varying radioiodine activity levels and different premedication regimens, particularly thyreostatic drugs and thyroid-blocking agents.

## 2. Materials and Methods

### 2.1. Animals

Female BALB/c mice (Charles River Laboratories Germany GmbH, Sandhofer Weg 797633 Sulzfeld, Germany, average weight: 25 g) were used in this study. The choice of female BALB/c mice was based on their consistent thyroid function, which is crucial for accurately studying radioiodine metabolism and exhalation patterns. Female BALB/c mice are well established in radiopharmaceutical research due to their stable thyroid hormone levels and reduced variability in iodine uptake [19]. Their reduced aggression and ease of handling further ensure minimal stress, enhancing the reproducibility and ethical conduct of the study [20,21,22]. The number of animals was minimized to comply with ethical standards. Approval for animal experiments was granted by the Cologne District Government (Approval No. K19, 42/00), and all protocols were coordinated with the university’s Animal Welfare Officer.

Mice were housed under standardized conditions (18–23 °C, artificial lighting) and provided with specialized feed and water ad libitum. They were kept in groups of up to five or ten, depending on cage size, and were regularly supplied with fresh water, food, and bedding.

### 2.2. Air Sampling and Measurements

The collection of air samples has been thoroughly detailed in previous publications by our research group [14,15]. Briefly, a “Portable Constant Flow Airsampler” (Model AVS-28 A, SAIC/RADeCO, RADeCO Inc. 17 West Parkway Plainfield, CT 06374, USA) was used to draw air from metabolic cages through a series of specialized filters (Figure 1).

The efficiency of radioiodine retention by the various filter materials (excluding glass fiber filters) was independently tested for each batch by Environmental Engineering & Testing, Inc. (Columbia Energy and Environmental Services 1806 Terminal Drive, Richland, WA 99354, USA) under contract with the American manufacturer (SAIC).

The activity of each filter was measured using a NaI(Tl) well-type detector (ISOMED 100, Nuklear-Medizintechnik Dresden GmbH, Rudolf-Renner-Str. 2, D-01109 Dresden, Germany). The total activity was determined as the sum of the activities of the individual filters.

Measurements in the well-type detector, analyzed using UMS software, version 4.40. on a PC workstation, were recorded in counts per second (cps). A conversion factor (cps to Bq) for the detector was established by using the known activity of an I-131-NaI solution (MDS Nordion, Avenue de l'Espérance 1, 6220 Fleurus, Belgium, 500 µL), which was evenly distributed over the surface of a filter in the same geometric configuration.

### 2.3. Animal Experiments: Experimental Setup

We provide a detailed description of the animal experiments conducted in this study, as this has not been published previously:

The mice were housed in metabolic cages (Süsse & Schmidt KG, Kasseler Straße 76, 34281 Gudensberg Germany) (Figure 2) during the measurement of I-131 concentration in exhaled air. To prevent contamination-related measurement inaccuracies, the metabolic cages were thoroughly cleaned before each measurement. The cages featured a conically shaped waste collection system, enabling the separate collection of feces and urine in a container beneath the cage during the 15 h measurement period. This design effectively shielded excreta, minimizing its impact on exhaled air measurements.

The water bottle and two food cylinders were placed in consistent, predefined positions for each mouse, ensuring a standardized setup and enhancing comparability. Two metabolic cages were operated in parallel, allowing for simultaneous individual measurements on two separate mice.

Each cage was equipped with a filter cartridge designed to hold an aerosol filter and four circular filters. The cartridge was loaded in a specific sequence, as demonstrated in Figure 1. Air from the metabolic cages was continuously drawn through the filtration system for the entire 15 h period using the “Portable Constant Flow Airsampler”. The airflow rate was approximately 4.5 m^3^/h.

### 2.4. Application of Radioiodine

The administration of various substances was performed orally using a button-tipped cannula (1.00 × 50 mm Luer, Hero Berlin GmbH, Schlesische Str. 26, 10997 Berlin, Germany).

Prior to administration, the mice received approximately 100 µL of an anesthetic mixture, consisting of 90 µL Ketanest^®^ (PFIZER PHARMA GmbH, Friedrichstraße 110, 10117 Berlin, Germany, 4 mg/25 mL) and 10 µL of a 2% Rompun^®^ (Bayer Vital GmbH, Kaiser-Wilhelm-Allee 1, 51368 Leverkusen, Germany) solution, which was administered intramuscularly. Subsequently, the button-tipped cannula was orally inserted into the esophagus of each mouse, reaching just before the gastric entrance. The substances were then directly administered into the stomach via the cannula.

However, the exact positioning of the cannula could not be verified during the procedure, making it possible that some substances were delivered only to the esophagus rather than directly into the stomach. Additionally, over-insertion of the cannula, with potential perforation of the stomach wall, could not be entirely ruled out.

A relatively small volume of approximately 100 µL of the anesthetic mixture was sufficient to keep the mice adequately sedated for substance administration.

### 2.5. Relationship Between Administered Activity and I-131 Exhalation

In the first series of experiments, the dose dependence of radioiodine exhalation was investigated. Six mice, under the previously described anesthesia, were orally administered defined amounts of I-131. The administered activities were 0.1 MBq, 1 MBq, 10 MBq, and 23 MBq, corresponding to 5 to 1150 MBq/kg, based on an average body weight of 20 g.

Activities of 0.1 MBq and 1 MBq fall within the typical dosage range for radioiodine therapy in humans, whereas 10 MBq and 23 MBq exceed the therapeutic range. To achieve the desired activity, radioiodine was diluted with water to a final volume of approximately 200 µL per dose.

The exact administered activity was determined by measuring the filled cannula and syringe in an isotope calibrator before and after administration, with the difference representing the applied activity.

After dosing, each mouse was individually housed in a metabolic cage for 15 h, with access to food and water. During this period, cage air was continuously drawn through a filtration system.

### 2.6. Pre-Application of Substances Affecting Radioiodine Retention or Iodine Metabolism

Control group mice received [^131^I]I-iodide (0.1 MBq/animal or 10 MBq/animal) without any pre-application treatment.

The experimental procedure varied slightly depending on the type of pre-application treatment. All pre-application protocols involved anesthetizing the mice with the previously described mixture of Ketanest S and Rompun. Administration was performed orally using a button-tipped cannula, ensuring that the substance was delivered directly into the stomach. For all test series, the administered volume was 200 µL.

For animals with thyroid function blocked by varying amounts of cold iodine (Merck KGaA, Frankfurter Str. 250, 64293 Darmstadt, Germany) or perchlorate (Bayer Vital GmbH, Kaiser-Wilhelm-Allee 1, 51368 Leverkusen Germany), administration occurred one hour before administering 0.1 MBq/animal or 10 MBq/animal of radioiodine.

Carbimazole and L-thyroxine (Henning Berlin GmbH, Komturstr. 58-62, 12099 Berlin, Germany) were administered daily for ten days. Given the known impact of food intake on L-thyroxine absorption, these mice were fasted for at least six hours before each daily dose. On the eleventh day, [^131^I]I-iodide was administered. The administration followed a standardized protocol, as previously described (button-tipped cannula inserted into the stomach, volume: 200 µL).

For animals with thyroid glands blocked by cold iodine or perchlorate, administration was performed one hour before the radioiodine dose.

### 2.7. Statistics

For the statistical analysis, separate two-way ANOVAs were conducted for each activity level (e.g., 0.1 MBq and 10 MBq) and for each pre-applied substance (e.g., KI, KClO_4_, carbimazole, and L-thyroxine) (GraphPad Prism 10.4.1, GraphPad Software, LLC, 1608 Solana Beach, Suite 100, San Diego, CA 92075, USA).

The dependent variable was the percentage composition of each I-131 species. The ANOVA allowed for the assessment of the main effects of each factor (dosage and I-131 species type) and any interaction effects, indicating whether the impact of dosage varied according to the type of exhaled species. Statistical significance was set at *p* < 0.05. When significant main effects or interactions were observed, a Tukey post hoc test was applied to identify specific significant differences among dosages and I-131 species types.

One-way ANOVA analyses were performed to evaluate whether premedications with potassium iodide (KI), perchlorate (KClO_4_), L-thyroxine, and carbimazole influenced the total activity of exhaled I-131. For each experimental condition, the total exhaled activity was calculated as the mean of individual measurements, with standard deviations and sample sizes recorded for statistical input.

The data were grouped based on premedication type and dose, and an ANOVA was conducted to determine significant differences among the groups. Post hoc multiple comparisons tests were used to identify specific group differences when the ANOVA results indicated statistical significance. This approach provided a robust assessment of the impact of pharmacological interventions on I-131 exhalation.

## 3. Results

### 3.1. Separation Efficiency of the Filter Combination

The separation efficiency analysis of the filter combination, as described above, is summarized in Table 1. The filtration system demonstrated high efficacy in differentiating between aerosolized, molecular, and organically bound iodine. The I-131 radioactivity collected in the final filter (TEDA-impregnated activated charcoal) represents a mixture of molecular and organically bound iodine that was not retained by the preceding filtration stages. The filtration setup effectively retained aerosolized and molecular iodine, while organically bound iodine was partially captured in intermediate filtration stages. The final TEDA-impregnated activated charcoal filter collected residual molecular and organically bound iodine, indicating that a fraction of these species was not retained by the preceding filters.

### 3.2. Investigation of the Extent of I-131 Exhalation and Its Chemical Composition in the Breath of Mice Following Administration of Increasing I-131 Radioactivities

Figure 3 depicts the linear regression analysis between orally administered I-131 activity and exhaled I-131, demonstrating a positive correlation, where higher administered activity corresponds to increased exhaled radioiodine levels.

Table 2 presents the percentage of exhaled I-131 relative to the administered I-131 activity in the breath of mice, along with the corresponding standard deviations and sample sizes for each activity level.

The results in Figure 3 demonstrate a linear relationship between administered I-131 activity and exhaled I-131 in mice (R^2^ = 0.9819), indicating that 98.2% of the variability in exhaled I-131 can be attributed to the administered dose. The statistically significant slope (*p* < 0.0001) highlights a consistent dose–response effect, where each increase in administered I-131 leads to a proportional rise in exhaled activity. This relationship spans a broad range of administered activities, reinforcing the predictive strength of the model.

Furthermore, as shown in Table 2, the percentage of exhaled I-131 relative to the administered dose remains constant across different activity levels. Despite substantial variations in administered activity, the exhaled I-131 consistently falls within the range of 0.2% to 0.3%.

Figure 4 depicts the percentage composition of exhaled I-131 species (aerosol, elemental I_2_, and organically bound I-131) as a function of administered I-131 activity, illustrating the changes in species distribution with increasing I-131 doses.

The data indicate that organically bound I-131 consistently constitutes the largest fraction of exhaled I-131, while aerosolized and molecular iodine (I_2_) remain lower and relatively stable across all activity levels. No significant dose-dependent increase in the proportion of organically bound I-131 was observed.

The two-way ANOVA revealed that, at 0.1 MBq, there was no significant difference between the aerosol and I_2_ fractions (*p* = 0.8780). However, when comparing either of these fractions with the organically bound I-131 fraction, highly significant differences emerged (*p* < 0.0001 for both aerosol vs. organically bound I-131 and I_2_ vs. organically bound I-131). This pattern remained consistent as the administered activity increased, underscoring the strong influence of treatment on the composition of exhaled I-131.

As the activity increased to 10 MBq, the difference between the aerosol and I_2_ fractions remained statistically insignificant (*p* = 0.7324), indicating that these two fractions exhibit relative stability across different activity levels. In contrast, the comparisons involving the organically bound I-131 fraction continued to demonstrate statistically significant differences. This trend persisted at 23 MBq, where significant differences were observed between aerosol, I₂, and organically bound I-131 fractions.

These findings indicate that the proportion of organically bound I-131 does not depend on the administered I-131 activity level. Despite significant variations in the administered activity, the organically bound fraction remains virtually unchanged.

### 3.3. Impact of Pre-Application of Stable Potassium Iodide (0.01–1 mg per Animal) on Chemical Composition of Exhaled Iodine in the Breath of Mice

Figure 5 illustrates the percentage composition of exhaled I-131 species (aerosol, elemental I_2_, and organically bound I-131) in mice as a function of co-administered potassium iodide (KI) doses, comparing the effects of 0.1 MBq and 10 MBq of administered I-131 activity.

This figure reveals a clear trend following the oral administration of 0.1 MBq and 10 MBq of I-131: as potassium iodide (KI) doses increase, the proportion of organically bound I-131 decreases, while the proportion of elemental iodine (I_2_) increases. The aerosol fraction remains unchanged, exhibiting no significant variation in response to either the pre-applied KI dose or the administered I-131 activity level.

The two-way ANOVA confirms that the composition of exhaled I-131 species is significantly influenced by KI dose (*p* < 0.0001). The Tukey multiple comparisons test was applied to assess the statistical significance of differences between species types within each KI dose group at both 0.1 MBq and 10 MBq of orally administered I-131.

In the 0.1 MBq activity group, a significant increase in the proportion of elemental I-131 was observed compared to the control group at all pre-applied KI doses, particularly at 1 mg and 5 mg of KI (0.01 mg: *p* = 0.0188, 0.1 mg: *p* = 0.0118, 1 mg: *p* < 0.0001). Additionally, significant differences were detected between 0.01 mg and 1 mg (*p* < 0.0001) and between 0.1 mg and 1 mg (*p* < 0.0001) of pre-applied KI within the 0.1 MBq group. Conversely, the proportion of organically bound iodine exhibited a significant decrease with increasing doses of pre-applied potassium iodide, compared to the control group (0.01 mg: *p* = 0.0078, 0.1 mg: *p* = 0.0006, 1 mg: *p* < 0.0001).

In the 10 MBq group, a significant increase in elemental iodine in exhaled breath was observed across all doses of pre-applied potassium iodide (KI) compared to the control group (*p* < 0.0001). A similar effect was noted when the pre-applied KI dose increased from 0.01 mg to 0.1 mg (*p* < 0.0001) and from 0.01 mg to 1 mg (*p* = 0.0127). However, increasing KI from 0.1 mg to 1 mg resulted in a slight but non-significant reversal of this trend.

The impact of pre-applied KI on the percentage of organically bound iodine showed a consistent and significant decrease across all KI doses, compared to the control group (*p* < 0.0001). Increasing the pre-applied KI dose from 0.01 mg to 0.1 mg also resulted in a significant reduction in the organic radioiodine fraction (*p* = 0.0017). Increasing KI from 0.01 mg to 1 mg or from 0.1 mg to 1 mg showed a non-significant trend toward a KI-dependent reduction in the percentage of organically bound radioiodine.

When comparing the same pre-applied KI doses between the 0.1 MBq and 10 MBq groups, no significant differences were observed in the aerosol fraction between the two activity groups. However, following the pre-application of 0.01 mg and 0.1 mg of KI, the 10 MBq group exhibited a significant increase in the proportion of radioiodine exhaled in elemental form compared to the 0.1 MBq group (0.01 mg KI: *p* = 0.0194; 0.1 mg KI: *p* < 0.0001).

In contrast, the proportion of organically bound iodine showed a significant decrease with 0.1 mg KI (0.01 mg KI: *p* = 0.079, not significant; 0.1 mg KI: *p* < 0.0001). The results with 1 mg KI pre-application were less conclusive, with no significant differences detected.

### 3.4. Impact of Pre-Application of Sodium Perchlorate (0.5–5 mg per Animal) on the Chemical Composition of Exhaled Iodine in the Breath of Mice

Figure 6 illustrates the percentage composition of exhaled I-131 species (aerosol, elemental I_2_, and organically bound I-131) in mice as a function of co-administered potassium iodide (KI) doses, comparing the effects of 0.1 MBq and 10 MBq of administered I-131 activity.

With increasing potassium perchlorate (KClO_4_) doses from 0.05 mg to 5 mg, a clear trend is particularly evident in the 10 MBq group: the proportion of organically bound I-131 decreases, while the proportion of elemental iodine (I_2_) increases (Figure 6). In contrast, this trend is less consistent in the 0.1 MBq group. The aerosol fraction remains unchanged across different KClO_4_ doses, showing no significant variation in either the 0.1 MBq or 10 MBq activity group.

The two-way ANOVA confirms that the KClO_4_ dose has a highly significant effect on the composition of exhaled I-131 species (*p* < 0.0001). The Tukey multiple comparisons test was applied to assess the statistical significance of differences between species types within each KClO_4_ dose group for both 0.1 MBq and 10 MBq of orally administered I-131.

In the 0.1 MBq group, a significant increase in the proportion of elemental I-131 was observed compared to the control group at 1 mg and 5 mg of pre-applied KClO_4_ (*p* < 0.0001). Similarly, the proportion of organically bound I-131 showed a significant decrease (*p* < 0.0001). Statistically significant differences were also detected between 0.05 mg and 1 mg (*p* = 0.0002) and between 0.05 mg and 5 mg (*p* = 0.0004) of pre-applied perchlorate. Increasing KClO_4_ from 1 mg to 5 mg resulted in a slight decrease in the proportion of exhaled organically bound I-131, deviating from the expected trend, but this difference was not statistically significant.

In the 10 MBq group, a clear dose-dependent trend was observed: with increasing pre-applied KClO_4_ doses, the proportion of elemental I-131 increased, while the proportion of organically bound I-131 decreased. However, Tukey’s multiple comparisons test revealed that differences in exhaled radioiodine composition due to pre-applied KClO_4_ doses were not statistically significant, even compared to the control group, except for elemental I-131, where a significant difference was observed between the control and 5 mg KClO_4_ group (*p* < 0.0001).

No significant influence of increasing orally administered I-131 activity on the chemical composition of exhaled radioiodine was detected for the same pre-applied KClO_4_ dose.

### 3.5. Impact of Pre-Application of Carbimazole (0.2 mg over 10 Days) and L-Thyroxine (0.3 mg over 10 Days) on the Chemical Composition of Exhaled Iodine in the Breath of Mice

Figure 7 illustrates the percentage composition of exhaled I-131 species (aerosol, elemental I_2_, and organically bound I-131) in mice as influenced by the co-administration of carbimazole or L-thyroxine.

In the 0.1 MBq group, premedication with carbimazole or L-thyroxine had no significant effect on the percentage composition of exhaled radioiodine species in mice.

In the 10 MBq group, however, a clear trend emerged, distinguishing this premedication from previous pre-applications with KI or KClO_4_. Specifically, there was a notable increase in the percentage of I-131 exhaled as an aerosol. The effect of carbimazole was borderline significant compared to the control group (*p* = 0.052), whereas the effect of L-thyroxine was highly significant (*p* < 0.0001).

This increase in aerosol formation appears to occur at the expense of the organically bound iodine fraction. Both carbimazole (*p* = 0.0021) and L-thyroxine (*p* = 0.001) induced a significant decrease in the percentage of organically bound iodine in exhaled breath.

### 3.6. Influence of Exogenous Modulations on the Total Amounts of Exhaled I-131

The results are summarized in Table 3.

In a few cases, a statistically significant influence of premedications on the total amount of exhaled I-131 was observed. For an administered activity of 0.1 MBq, the effects were less pronounced compared to 10 MBq (Table 3), and in most cases, the standard deviation was too high to draw definitive conclusions.

A substantial effect was observed with potassium iodide (KI), where the total amount of exhaled radioiodine increased with higher pre-administration doses of “cold” iodine. Tukey’s multiple comparisons test revealed a significant difference compared to the control group at a premedication dose of 0.1 mg or higher (*p* = 0.0361). The 1 mg KI premedication showed a near-significant trend (*p* = 0.0536).

Additionally, carbimazole premedication significantly reduced the total amount of exhaled I-131 following oral administration of 10 MBq compared to controls (*p* = 0.0019). L-thyroxine also exhibited a tendency toward reducing exhaled radioiodine activity, though this effect did not reach statistical significance.

At the lower dose of 0.1 MBq of orally administered I-131, no significant influence of premedications on the total amount of exhaled I-131 was detected. Similarly, pre-administration of perchlorate (KClO_4_) did not result in a statistically significant effect on the total amount of exhaled iodine.

This study provides insights into the mechanisms of radioiodine exhalation in mice following oral administration of I-131 and the effects of various pharmacological interventions.

## 4. Discussion

### 4.1. Alignment with Existing Clinical Investigations on Radioiodine Exhalation

The exhalation of radioiodine following radioiodine therapy (RIT) has been well documented in various studies, both in the ambient air of patient rooms within nuclear medicine therapy units and in the exhaled breath of patients after RIT [16,17,18]. These investigations have demonstrated that radioiodine is predominantly exhaled in an organically bound form, while a smaller fraction is present as elemental iodine (I_2_) and aerosolized iodine species. Breath analysis studies in patients undergoing RIT have indicated that exhaled radioiodine typically constitutes 0.01–0.1% of the administered activity, with significant interindividual variability [18]. The predominance of organically bound iodine in exhaled breath is a key finding that underscores the metabolic complexity of iodine handling in vivo.

Our study in mice aligns with these human findings by confirming that organically bound iodine is the major fraction of exhaled radioiodine across all tested conditions. Notably, we observed that the exhaled fraction in mice (~0.2–0.3%) was an order of magnitude higher than in humans, consistent with the significantly higher metabolic rate of rodents. Female mice were specifically selected for this study due to their established reliability in thyroid-related investigations and their generally more docile nature [19,20,21,22]. Furthermore, the proportion of exhaled organic iodine remained stable across different administered activities (0.1–23 MBq). This finding merely indicates that the administered radioiodine activity has no impact on the chemical composition of exhaled radioiodine. The resulting increase in the absolute amount of substance (theoretically, 23 MBq I-131 equals 5 ng iodine) remains entirely negligible, as the quantity of iodine introduced solely through radioactivity is far too small to exert any influence on the chemical composition of exhaled iodine.

### 4.2. Mechanisms of Iodine Oxidation to I_2_ and Exhalation

Iodide oxidation to molecular iodine (I_2_) occurs primarily in the thyroid gland, where thyroid peroxidase (TPO) catalyzes the reaction in the presence of hydrogen peroxide (H_2_O_2_) [23,24]. Elemental iodine in small amounts will be dissolved in water (aqueous iodine solution, “iodine water”) [25,26]. We assume that the transfer of [^131^I]-iodine water into exhaled air likely occurs via passive diffusion across the alveolar–capillary membrane, driven by its partial pressure gradient between blood and the pulmonary airspace.

However, comparable oxidation processes are also present in several extrathyroidal tissues, including the gastric mucosa and lungs, where lactoperoxidase (LPO) facilitates iodine oxidation in secretions [27,28]. This suggests multiple systemic pathways for iodine oxidation, supporting the observed presence of I_2_ and organic iodine species in exhaled breath.

Our findings may further be explained by assuming that iodine exhalation is influenced not only by enzymatic oxidation but also by radiation-induced radical formation. Beta and gamma radiation from I-131 generates reactive iodine radicals (I·), which can undergo secondary recombination to form I_2_ [29,30,31,32,33]. These reactions are facilitated by reactive oxygen species (ROSs), which contribute to the oxidation of iodide into I_2_ [34,35] suitable for pulmonary elimination.

### 4.3. Formation and Pulmonary Elimination of Organically Bound Iodine

A key observation in both human and murine studies is the persistence of exhaled organic iodine despite complete thyroid blockade. This suggests the presence of alternative iodine metabolism pathways beyond classical thyroidal organification. Our data strongly support the hypothesis that methyl iodide (CH_3_I) is the dominant exhaled species, formed through the following potential mechanisms:Enzymatic Methylation:

The methylation of iodide (I^−^) via S-adenosylmethionine (SAM)-dependent methyltransferases is well known in microbial and hepatic metabolism. The involvement of gut microbiota or hepatic processing could explain systemic CH_3_I formation and subsequent pulmonary elimination [36,37].

2.Methylation Following Radiolysis by Reactive Radicals:

Beta decay from I-131 induces methyl radical (·CH_3_) formation, which promptly reacts with iodine radicals (I·) to form CH_3_I. This radiation-induced methylation process explains why organically bound iodine appears in exhaled breath even in the absence of thyroid function [38,39,40,41].

3.Microbial Contribution:

The gut microbiome actively metabolizes halides, including iodine, into methylated derivatives. Given the enterohepatic circulation of iodine, microbial methylation could contribute to CH_3_I formation, which enters systemic circulation and reaches the lungs for exhalation [42,43].

4.Hepatic and Pulmonary Metabolism of Iodine:

The liver, a known site for iodine metabolism, may facilitate the conjugation and demethylation of iodine-containing metabolites, allowing CH_3_I to enter circulation and be exhaled. The pulmonary system also plays a role in metabolizing and eliminating volatile iodine species, akin to other halogenated compounds.

Given the high volatility of CH_3_I and lipophilicity, it diffuses efficiently from blood into alveolar air, following concentration gradients similar to those observed with other volatile organic compounds (volatile radioactive iodine species, VOIs).

### 4.4. Impact of Thyroid-Blocking Agents on Radioiodine Exhalation

Pharmacological thyroid blockade with potassium iodide (KI) or perchlorate (KClO_4_) significantly altered radioiodine exhalation profiles. While these agents effectively reduced thyroidal iodine uptake, they increased the proportion of exhaled elemental iodine (I_2_) and organic iodine species, suggesting that iodine, when not sequestered by the thyroid, follows alternative metabolic routes.

These results further reinforce the idea that iodine metabolism extends beyond the thyroid, with exhalation driven by redistribution to extrathyroidal tissues and radical-mediated transformations in systemic circulation.

### 4.5. Effects of Antithyroid Drugs on Iodine Exhalation

The administration of carbimazole and L-thyroxine influenced iodine metabolism differently than thyroid-blocking agents. At higher administered activities (10 MBq/mouse), both drugs increased the formation of I-131-containing aerosols, likely due to the following:Enhanced radiation-induced radical formation in extrathyroidal tissues.Reduced thyroidal sequestration of iodine, allowing for greater systemic availability and alternative metabolism.Increased oxidative stress, leading to iodine-containing vesicle formation and pulmonary excretion.

This observation supports the idea that volatile iodine formation is not solely dictated by thyroidal hormone synthesis but is strongly influenced by extrathyroidal radical-driven reactions.

### 4.6. Future Research Directions

Quantitative analysis of VOI formation in exhaled breath using advanced spectrometric methods.Investigation of the gut microbiota’s role in iodine methylation and their contribution to systemic iodine metabolism.Radiation-driven iodine radical chemistry in extrathyroidal tissues, focusing on oxidative stress pathways and their implications for radioiodine therapy.

## 5. Conclusions

This study provides mechanistic perspectives on radioiodine exhalation and metabolism, demonstrating the following:Iodine exhalation is predominantly driven by extrathyroidal pathways, particularly when thyroidal uptake is blocked.Methyl iodide (CH_3_I) is the most likely exhaled organic iodine species, formed via enzymatic, microbial, and radiolytic pathways.Beta radiation-induced radical chemistry plays a significant role in volatile iodine formation, independent of thyroidal enzyme activity.The pharmacological modulation of thyroid function significantly alters exhalation patterns, emphasizing the systemic impact of iodine metabolism beyond traditional thyroidal pathways.

The insights presented here appear to be helpful concerning our understanding of radioiodine metabolism both in the thyroid and beyond the thyroid’s processes, which have been largely unknown up until now. Implications for radiation safety, dosimetry, and environmental iodine dynamics have to be considered. Future studies should aim to unravel biochemical pathways in more detail in order to better understand biochemical iodine species, especially iodine organification (e.g., methylation), and their relevance in clinical and environmental contexts.

## Figures and Tables

**Figure 1 biomedicines-13-00897-f001:**
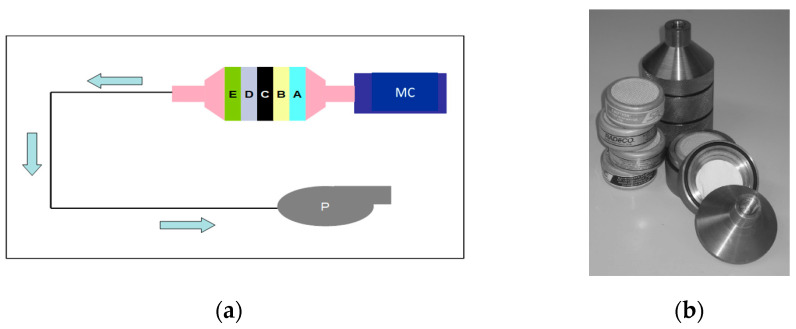
Schemes follow another format. If there are multiple panels, they should be listed as follows: (**a**) principle of measuring the percentage of different radioiodine species in the air (MC: metabolic cage, P: pump, A: glass fiber filter, B + C: cadmium iodide filters, D: silver zeolite filter: organically bound radioiodine, E: TEDA-impregnated activated charcoal: all radioiodine forms); (**b**) photograph of the filter housing and filters.

**Figure 2 biomedicines-13-00897-f002:**
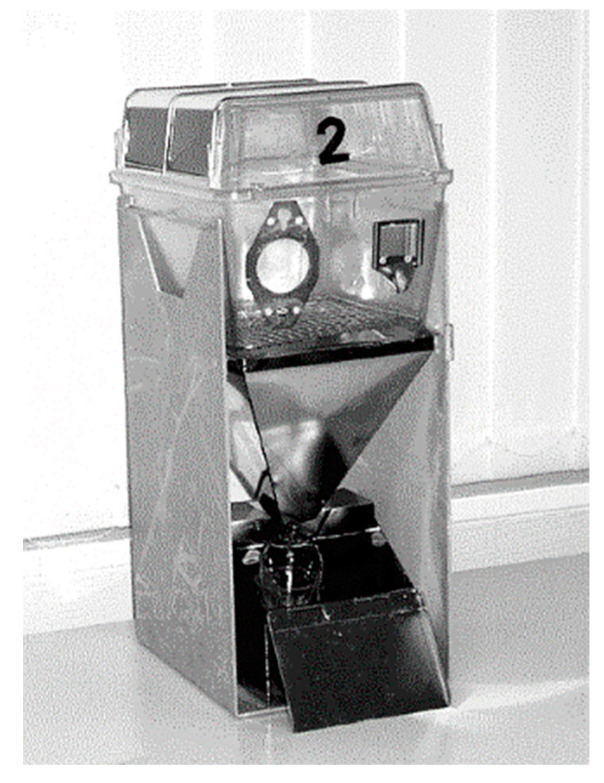
Metabolic cage for the studies.

**Figure 3 biomedicines-13-00897-f003:**
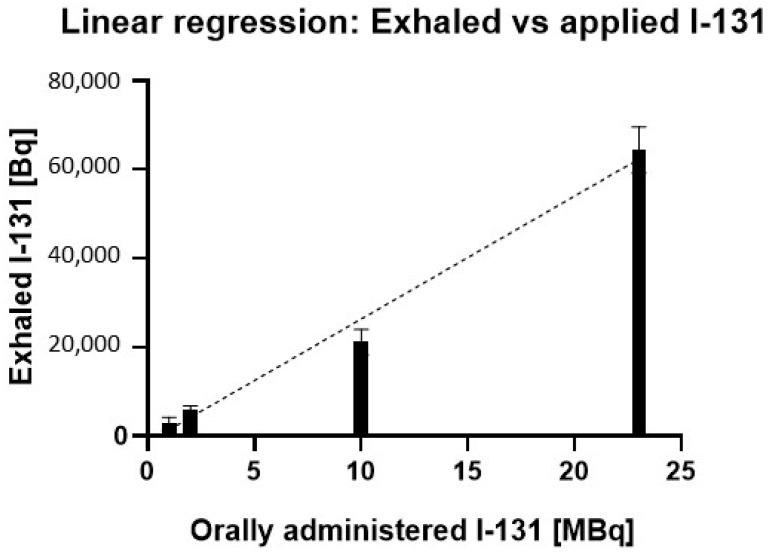
Linear relationship between orally administered I-131 activity and exhaled I-131 in mice. The regression analysis shows a strong positive correlation (R^2^ = 0.9819) between the administered I-131 dose (MBq) and the amount of I-131 exhaled (Bq). Error bars represent the standard error of the mean for each data point.

**Figure 4 biomedicines-13-00897-f004:**
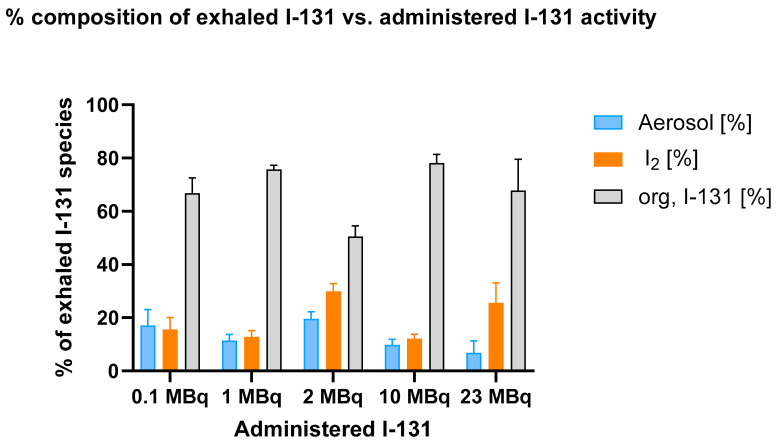
Percentage composition of exhaled I-131 species (aerosol, elemental I-131 (I_2_), and organically bound I-131) across different administered I-131 activities. The breath sample collection from the mice was conducted over a period of 15 h. Error bars represent the standard error of the mean for each data point.

**Figure 5 biomedicines-13-00897-f005:**
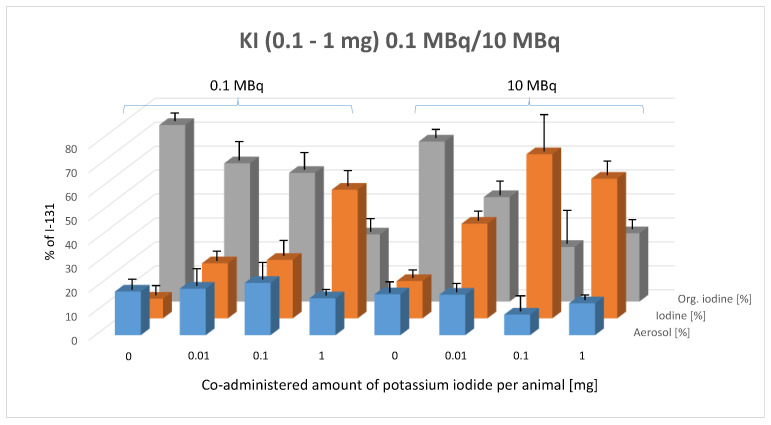
Effect of pre-administered potassium iodide (0.1–1 mg, given 1 h prior to radioiodine administration) on the composition of exhaled I-131 species (aerosol, elemental I-131 (I_2_), and organically bound I-131) at two I-131 activity levels (0.1 MBq and 10 MBq). Error bars represent the standard error of the mean for each data point.

**Figure 6 biomedicines-13-00897-f006:**
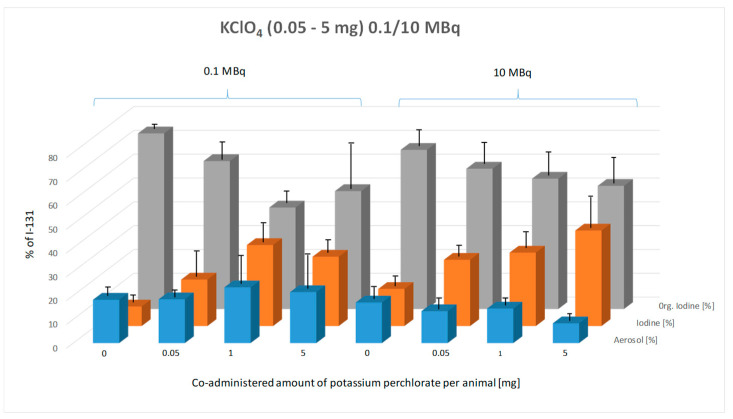
Effect of pre-administered potassium perchlorate (0.05–5 mg, given 1 h prior to radioiodine administration) on the composition of exhaled I-131 species (aerosol, elemental I-131 (I_2_), and organically bound I-131) at two administered I-131 activity levels (0.1 MBq and 10 MBq). Breath samples were collected from the mice over a 15 h period. The graph illustrates the percentage composition of each exhaled species across control and potassium perchlorate-treated groups. Error bars represent the standard error of the mean for each data point.

**Figure 7 biomedicines-13-00897-f007:**
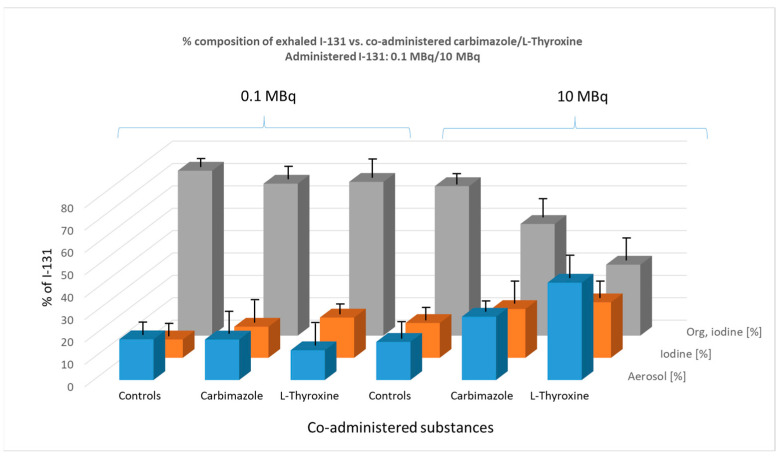
Effect of co-administered carbimazole (0.3 µg) and L-thyroxine (0.2 µg), administered daily over 10 days, on the composition of exhaled I-131 species (aerosol, elemental I-131 (I_2_), and organically bound I-131) at two administered I-131 activity levels (0.1 MBq and 10 MBq). The graph illustrates the percentage composition of each exhaled species. Error bars represent the standard error of the mean for each data point.

**Table 1 biomedicines-13-00897-t001:** Separation efficiency analysis for the filter combination.

Vapor Phase *	Fraction of Total I-131 Radioactivity in Filters in %					
	Glass Fiber Filter	CdI_2_ Filter I	CdI_2_ Filter II	Silver Zeolite Filter	TEDA-Activated Charcoal Filter	Glass Fiber Filter
I_2_	0.2	95.4	0.8	1.3	1.6	0.2
Ethyl iodide	0.6	1.0	1.0	96.9	0.9	0.6
Combined Vapor Phases ^*^	1.0	14.8	1.0	77.3	2.6	1.0

^*^ Radioactivity measurements of the ethanolic and aqueous phases showed an I_2_/ethyl iodide ratio of 1:4.5.

**Table 2 biomedicines-13-00897-t002:** Exhaled I-131 relative to applied I-131 in the breath of mice in %.

Administered Activity [MBq]	Exhaled I-131	Standard Deviation (SD)	N (Sample Size)
0.1 MBq	0.9	0.6	6
1 MBq	0.3	0.2	6
10 MBq	0.2	0.1	6
23 MBq	0.3	0.1	6

**Table 3 biomedicines-13-00897-t003:** Influence of exogenous modulations on the total amounts of exhaled I-131.

	0.1 MBq I-131			10 MBq I-131		
	Mean ^1^ [Bq]	Std. Dev.	N	Mean ^1^ [Bq]	Std. Dev.	N
Controls	214.0	102.8	4	21,150	2690	6
0.01 mg KI	351.3	243.8	6	22,882	7989	4
0.1 mg KI	297.6	234.0	5	45,490	21,167	5
1 mg KI	450.0	253.8	6	45,322	13,884	4
0.5 mg perchl.	254.5	187.0	8	22,665	5979	6
1 mg perchl.	406.3	131.8	8	26,802	10,938	6
5 mg perchl.	542.5	662.6	7	23,415	7291	6
L-thyroxine	161.0	147.6	6	11,277	10,698	6
Carbimazole	214.6	194.2	7	4954	2927	6

^1^ Mean I-131 activity measured across all filters over a 15 h period.

## Data Availability

The original contributions presented in this study are included in the article/Appendix A. Further inquiries can be directed to the corresponding author.

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
