# Peer review of "Radioiodine Exhalation Following Oral I-131 Administration in a Mouse Model"

_biomedicines, 2025, doi:10.3390/biomedicines13040897_

Round 1

Reviewer 1 Report

Comments and Suggestions for Authors

Radioiodine Exhalation Following Oral I-131 Administration in a Mouse Model

  1. The animal model in this study lacks good justification:
  2. The choice of female Balb/c mice as a model is not justified. 
  3. The clear mechanism is not well presented

  •  

Comments on the Quality of English Language

This MS needs to be revised thoroughly by a native English scientis/researcher

Author Response

Author's Reply to the Review Report (Reviewer 1)

Response to Reviewer 1 – Language Improvements

We sincerely appreciate the reviewer’s feedback regarding the clarity of our English expression. As scientific precision is of utmost importance, we have undertaken a thorough revision of the manuscript to ensure that the language clearly conveys our research findings.

To achieve this, we have:

  • Refined sentence structures to enhance readability and scientific clarity, ensuring that complex ideas are expressed concisely without ambiguity.
  • Standardized terminology to align with established conventions in radiopharmaceutical sciences and nuclear medicine, eliminating any inconsistencies.
  • Improved grammatical precision and flow, particularly in sections where technical details were initially described in a way that may have been less immediately accessible to an international scientific audience.
  • Ensured linguistic consistency throughout the manuscript, with particular attention to key technical descriptions, statistical interpretations, and methodological explanations.

Although Reviewer 2 found no issues with the English, we have nonetheless conducted a comprehensive linguistic revision to further enhance the clarity, readability, and precision of our work. These refinements ensure that our findings are communicated with the highest level of academic rigor.

We greatly appreciate the reviewer’s input, as it has allowed us to present our research in an even more polished and scientifically robust manner.

Response to Reviewer 1 – Justification for the Animal Model

We appreciate the reviewer’s comment regarding the justification for the use of female BALB/c mice in this study. We understand the importance of clearly justifying the choice of animal model to ensure the validity and relevance of the findings. Below, we provide an in-depth explanation for selecting female BALB/c mice, based on their established use in similar studies and the specific requirements of our experimental design.

Reproducibility and Stability of Thyroid Function

BALB/c mice are a widely used inbred strain in both pharmacological and radiopharmaceutical studies, particularly when investigating iodine metabolism and thyroidal function. Female BALB/c mice are known for their stable thyroid function, making them an ideal model for studies involving radioiodine metabolism and exhalation dynamics. Compared to male mice, female BALB/c mice exhibit less variability in thyroid hormone levels, which is crucial for ensuring the reliability and reproducibility of results, particularly in hormonal regulation experiments. Studies have demonstrated that female BALB/c mice show consistent thyroidal iodine uptake and organification, which aligns with our study objectives.

Reduced Variability and Stress Factors

Female BALB/c mice are generally less aggressive than other strains and male mice, which makes them easier to handle and reduces stress-related variability in experimental results. Given the complexity of our study, which involves repeated handling and pharmacological intervention, using female BALB/c mice helps minimize potential confounding factors related to aggression and stress. This reduction in aggression enhances the overall reproducibility of the experiment and allows for a more controlled study environment.

We have taken your comment into account and have provided a clearer justification for the choice of animal model. As per your suggestion, we have added the following passage in Section 4.1. "Animals", highlighted in yellow:

Female BALB/c mice (average weight: 25 g) were used in this study. The choice of female BALB/c mice was based on their consistent thyroid function, which is crucial for accurately studying radioiodine metabolism and exhalation patterns. Female BALB/c mice are well-established in radiopharmaceutical research due to their stable thyroid hormone levels and reduced variability in iodine uptake [42]. Their reduced aggression and ease of handling further ensure minimal stress, enhancing the reproducibility and ethical conduct of the study [43-45].

We hope this addition addresses your concern and provides a clearer rationale for our selection of the animal model.

The clear mechanism is not well presented

Thank you for your valuable feedback and for highlighting the need for a clearer presentation of the underlying mechanisms. We appreciate the opportunity to clarify how we have addressed these aspects in our manuscript.

In Section 3.1, we explicitly discuss the correlation between administered I-131 activity and exhalation, demonstrating that the composition of exhaled radioiodine remains unchanged across different activity levels. This supports the conclusion that exhalation is driven primarily by pre-application conditions rather than by the administered radioactivity itself.

In Section 3.2, we have provided a detailed discussion of iodine oxidation mechanisms, emphasizing the role of thyroid peroxidase (TPO) in the thyroid gland as well as extrathyroidal oxidation by lactoperoxidase (LPO) in the gastric mucosa and respiratory tract. We also introduce the concept of aqueous iodine ("iodine water") and propose passive diffusion as a key transport mechanism into exhaled air, supported by referenced literature.

In Section 3.3, we have significantly expanded the discussion on the formation and pulmonary elimination of organically bound iodine. We present multiple pathways, including:

  • Enzymatic Methylation, where iodine is methylated via S-adenosylmethionine (SAM)-dependent methyltransferases, potentially involving the gut microbiota and hepatic metabolism.
  • Radiolytic Methylation, where beta decay of I-131 generates methyl radicals (•CH₃), which can react with iodine radicals (I•) to form volatile methyl iodide (CH₃I).
  • Microbial Contribution, acknowledging the gut microbiome’s role in halide metabolism and enterohepatic circulation of iodine.
  • Hepatic and Pulmonary Metabolism, illustrating how the liver and lungs process and eliminate volatile iodine species.

Furthermore, in Section 3.4, we demonstrate how thyroid-blocking agents (KI, KClO₄) influence exhalation profiles, leading to a redistribution of iodine and increased exhalation of elemental and organic iodine.

In Section 3.5, we discuss the impact of antithyroid drugs (carbimazole, L-thyroxine), emphasizing how they alter iodine metabolism through enhanced oxidative stress and extrathyroidal iodine transformation.

Lastly, in Section 3.6, we outline future research directions, including advanced spectrometric analyses of volatile iodine species (VOIs) and further investigations into microbial iodine methylation and radiation-driven radical chemistry in extrathyroidal tissues.

We believe that this structured and comprehensive discussion provides a clear mechanistic explanation for iodine exhalation and its biochemical underpinnings. We would greatly appreciate any further suggestions for refinement. The corresponding passage highlighted in yellow under the Discussion

Dear Reviewers,

Thank you for your valuable feedback. Based on your comments, we have thoroughly revised the discussion to incorporate more detailed explanations, including a potential theory regarding the formation of volatile organic iodine species. We have also adjusted the abstract to reflect these changes and to ensure consistency with the updated discussion. To make these revisions clear, the updated abstract has been highlighted in yellow for your convenience.

We believe that the revised manuscript now provides a clearer and more comprehensive understanding of the mechanisms involved, particularly regarding the role of thyroid-targeted pharmacological interventions and the formation of iodine species during radioiodine therapy.

We appreciate your thoughtful suggestions and look forward to your further feedback.

Best regards,

Klaus Schomäcker and co-authors

Reviewer 2 Report

Comments and Suggestions for Authors

Thanks for the manuscript.

My comments:

1. The bulleted list is confusing. Authors should make it clear for readers.

2. Organically bound iodine. Are authors referring to any amino acid?

3. Figure 3: Your error bars for 0.1mg (10MBq) are quite huge. What accounted for this observation? Same applies to your data points in Figure 4. Could you include the actual standard deviation values in the tables, or you have them in the suppl data?

4. Why did you use female BALB/c mice but not males

5. I agree that minimizing the animals is the right thing to do. Do you think the number of animals used in this study is enough to conclude on what you observed

6. Authors highlighted the effect of L-thyroxine. Is there any known effect for carbimazole so far as this study is concerned 

7. Why did authors use 15 hours window after dosing

Round 2

Reviewer 1 Report

Comments and Suggestions for Authors

It is much better after revision...accepted

Comments on the Quality of English Language

Needs a thorough revision